# Variation in anthelmintic responses are driven by genetic differences among diverse *C. elegans* wild strains

**Amanda O. Shaver**[1], **Janneke Wit**[1], **Clayton M. Dilks**[1], **Timothy A. Crombie**[1], **Hanchen Li**[2], **Raffi V. Aroian**[2], **Erik C. Andersen**[1] *

**1** Molecular Biosciences, Northwestern University, Evanston, Illinois, United States of America, **2** Program in Molecular Medicine, University of Massachusetts Medical School, Worcester, Massachusetts, United States of America

* erik.andersen@gmail.com

**Data Availability Statement:** All code and data used to replicate the data analysis and figures are available at https://github.com/AndersenLab/anthelmintic_dose_responses_manuscript and

## Abstract

Treatment of parasitic nematode infections in humans and livestock relies on a limited arsenal of anthelmintic drugs that have historically reduced parasite burdens. However, anthelmintic resistance (AR) is increasing, and little is known about the molecular and genetic causes of resistance for most drugs. The free-living roundworm *Caenorhabditis elegans* has proven to be a tractable model to understand AR, where studies have led to the identification of molecular targets of all major anthelmintic drug classes. Here, we used genetically diverse *C. elegans* strains to perform dose-response analyses across 26 anthelmintic drugs that represent the three major anthelmintic drug classes (benzimidazoles, macrocyclic lactones, and nicotinic acetylcholine receptor agonists) in addition to seven other anthelmintic classes. First, we found that *C. elegans* strains displayed similar anthelmintic responses within drug classes and significant variation across drug classes. Next, we compared the effective concentration estimates to induce a 10% maximal response ($EC_{10}$) and slope estimates of each dose-response curve of each strain to the laboratory reference strain, which enabled the identification of anthelmintics with population-wide differences to understand how genetics contribute to AR. Because genetically diverse strains displayed differential susceptibilities within and across anthelmintics, we show that *C. elegans* is a useful model for screening potential nematicides before applications to helminths. Third, we quantified the levels of anthelmintic response variation caused by genetic differences among individuals (heritability) to each drug and observed a significant correlation between exposure closest to the $EC_{10}$ and the exposure that exhibited the most heritable responses. These results suggest drugs to prioritize in genome-wide association studies, which will enable the identification of AR genes.

## Author summary

Parasitic nematodes infect most animal species and significantly impact human and animal health. Control of parasitic nematodes in host species relies on a limited collection of

Zenodo at https://doi.org/10.5281/zenodo.7351693.

**Funding:** The authors received no specific funding for this work.

**Competing interests:** The authors have declared that no competing interests exist.

anthelmintic drugs. However, anthelmintic resistance is widespread, which threatens our ability to control parasitic nematode populations. Here, we used the non-parasitic roundworm *Caenorhabditis elegans* as a model to study anthelmintic resistance across 26 anthelmintics that span ten drug classes. We leveraged the genetic diversity of *C. elegans* to quantify anthelmintic responses across a range of doses, estimate dose-response curves, fit strain-specific model parameters, and calculate the contributions of genetics to these parameters. We found that genetic variation within a species plays a considerable role in anthelmintic responses within and across drug classes. Our results emphasize how the incorporation of genetically diverse *C. elegans* strains is necessary to understand anthelmintic response variation found in natural populations. These results highlight drugs to prioritize in future mapping studies to identify genes involved in anthelmintic resistance.

## Introduction

Parasitic nematodes are incredibly diverse and infect most animal and plant species [1,2]. Treatments rely on a limited arsenal of anthelmintic drugs with the same drug classes used across most parasite species [3]. The three major anthelmintic classes are benzimidazoles (BZs), macrocyclic lactones (MLs), and nicotinic acetylcholine receptor (nAChR) agonists. Over-reliance and inappropriate use of these anthelmintics have placed strong selective pressures on parasites and caused the evolution of anthelmintic resistance (AR) to every drug class [3]. In many cases, AR is highly heritable, which suggests that genetics plays an important role in the evolution of resistant nematodes. Therefore, calculating the heritability of AR allows us to estimate the fraction of phenotypic variation in anthelmintic responses that can be explained by genetic variation in a population [4]. The narrow-sense heritability is a measure of the genetic component that is contributed by alleles that act additively so it can be used to calculate how much AR will increase when drug selective pressure is applied to a population [4]. Because AR is highly heritable, we can identify the genes involved in AR and model resistance over time. These measures are critical because AR causes the drug to become ineffective [5]. With this knowledge, we can responsibly apply drugs and implement treatment strategies to reduce our global infection rate and burden of parasitic nematodes.

Most of our knowledge about mechanisms of resistance comes from studies of a single strain within a single species, the laboratory-adapted strain of *Caenorhabditis elegans* called N2. However, a single genetic background, whether in a free-living or parasitic species, cannot capture the enormous diversity present in the entire species, nor can it predict how natural populations of parasitic nematodes will respond to a drug [6]. In aggregate, single strains from many species might capture phylum-level variation, which strengthens the opportunity to identify genes involved in mechanisms of resistance. It is difficult to accurately test AR in genetically diverse parasitic nematodes because of multiple factors, including a lack of access to relevant life cycle stages, lack of global sample collections, host-dependent and cost-intensive laboratory life cycles, complex or non-existent *in vitro* culture systems, and a limited molecular toolkit [6].

With its ease of growth, genetic tractability, and ample molecular toolkit, the roundworm *C. elegans* is our most useful model to study AR. To date, *C. elegans* has contributed to the identification and characterization of mechanisms of resistance of all major anthelmintic drug classes [7–13]. Additionally, the natural genetic variation across the *C. elegans* species is accessible and continuously archived in the *C. elegans* Natural Diversity Resource (CeNDR), which has facilitated the characterization of natural responses to anthelmintic drugs [4,10,14].

Whole-genome sequence data and identified AR variants are available for all CeNDR strains, so orthologous genes between *C. elegans* and parasites can be queried to determine if *C. elegans* might be a good model for parasitic helminths [7]. Lastly, because of the tractability of *C. elegans* and established high-throughput assays (HTA), we can measure *C. elegans* responses to any soluble compound [15]. Thus, the genetic diversity of *C. elegans* can enable the discovery of the molecular targets of anthelmintics and has repeatedly proven to translate across parasitic nematode species [7,8,10,16–18].

Here, we performed dose-response analyses that used 26 anthelmintics across six genetically diverse *C. elegans* strains to identify how development was affected. The anthelmintics used in this study represent the three major anthelmintic drug classes (BZs, MLs, and nAChR agonists) in addition to seven other classes of anthelmintics (nicotinic acetylcholine receptor [nAChR] antagonists, a pore-forming crystal protein, a cyclicoctadepsipeptide, diethylcarbamazine, piperazine, a salicylanilide, and schistosomicides). We measured nematode development after drug exposure for six genetically diverse *C. elegans* strains using an established high-throughput phenotyping assay [15]. We assayed the strains with high levels of replication, collecting a total of 48,343 replicate anthelmintic responses across genetically diverse *C. elegans* strains, a throughput not possible using parasites. We used phenotypic responses to each anthelmintic to estimate dose-response curves, fit strain-specific model parameters, and calculate the contributions of genetics to these anthelmintic responses. Our results emphasize how the incorporation of natural genetic variation is necessary to quantify drug responses and identify the range of drug susceptibilities in natural populations. Importantly, studies focusing on genetic variation increase the likelihood of identifying orthologous genes between *C. elegans* and parasites of interest and, in turn, discover mechanisms of resistance shared across species [7].

## Results and discussion

### High-throughput assays across six wild strains facilitated dose-response assessments of 26 anthelmintic drugs

Dose-response assessments were performed using a microscopy-based high-throughput phenotyping assay for developmental delay in response to 26 anthelmintics (**Fig 1**). Anthelmintics assayed were five different BZs, seven MLs, three nAChR agonists, three nAChR antagonists, one pore-forming crystal toxin, one cyclooctadepsipeptide, one diethylcarbamazine, one pyrazinoisoquinoline, one salicylanilide, and three schistosomicides (**Table 1**). Six genetically diverse *C. elegans* strains were exposed to each anthelmintic in high replication. After measuring nematode responses, phenotypic data were cleaned and processed (see *Methods*). Next, dose-response curves were estimated for each anthelmintic to describe how genetic variation contributed to differences in anthelmintic resistance among strains. Differences in responses were measured by a change in developmental rate, as measured by animal length, a trait similar to larval development assay (LDA) [19]. Nematodes grow longer over time, and anthelmintics have been shown to slow this development [8,9,17,20–22]. Therefore, shorter animals after drug exposure demonstrated that the anthelmintic had a detrimental effect on development.

A four-parameter log-logistic dose-response curve was modeled for each of the 26 anthelmintics, where normalized median animal length was used as the metric for a phenotypic response (see *Methods*). For each strain-specific dose-response model, slope (*b*) and effective concentration (*e*) were estimated with strain as a covariate (**S1** and **S2 Tables**). $EC_{10}$ estimates were found to be more heritable than half maximal effective concentration ($EC_{50}$) estimates and were therefore used throughout our analyses (see *Methods*) (**S1 Fig**). Dose-response relationships described how different strains were affected at varying levels of anthelmintic exposure providing insights into how genetic differences impact anthelmintic susceptibility.

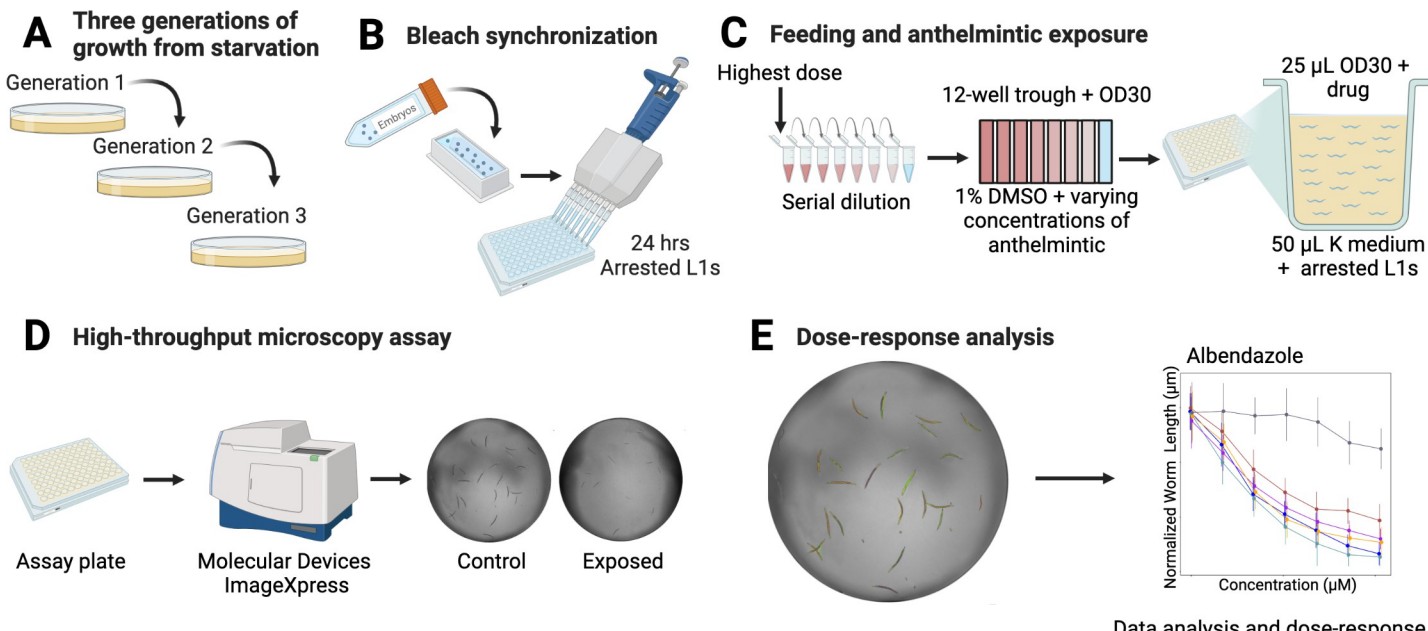

**Fig 1. The high-throughput phenotyping assay allows for rapid dose-response assessments across genetically diverse *C. elegans* strains. (A)** Strains were passaged for three generations to reduce generational effects of starvation. **(B)** Strains were bleach synchronized to collect embryos and then hatched and arrested at the L1 larval stage. **(C)** After 24 hours, a serial dilution containing eight concentrations for each anthelmintic was created. The serial dilution was added to an aliquot of *E. coli*. This mixture was fed to the nematodes. **(D)** After 48 hours of growth, animals were imaged to collect phenotypic measurements. **(E)** Data were cleaned, and dose-response analysis was performed. Detailed descriptions of all steps can be found in *Methods*. Created with BioRender.com. Modified from a previous version [23].

To test for differences in AR among the strains, we looked for differences in the strain-specific dose-response model parameters. We found differences in $EC_{10}$ values for 22 anthelmintics (**S1 Table**). Next, we focused on $EC_{10}$ comparisons between the reference strain N2 and all other strains (**S3 Table**). In total, we observed 44 instances across 22 compounds where at least one strain was significantly more resistant or sensitive than the reference strain N2 using $EC_{10}$ as a proxy (Student's t-test, Bonferroni correction; $p_{adj} < 0.05$). Because most studies in *C. elegans* AR have been conducted using the laboratory reference strain, N2, or mutant strains in the N2 genetic background, these results emphasize the importance of using genetically diverse individuals to understand drug responses. Furthermore, the observed frequency of strains with significantly greater anthelmintic sensitivity than the N2 strain was different than what is expected under the null expectation (see *Methods*; Fisher exact test; $p < 0.05$), which suggests that diverse *C. elegans* strains are not equally likely to be susceptible or resistant with respect to the commonly used N2 reference strain. The strain MY16 displayed the most sensitivity and resistance compared to the N2 strain, making up 31% and 63% of the cases, respectively.

Out of the 130 strain-specific slope comparisons with respect to the N2 strain, we observed 92 instances across the 26 compounds where a strain had a significantly different slope than the N2 strain (**S4 Table**). However, slope estimate comparisons between the N2 strain and every other strain only describe part of the breadth of *C. elegans* anthelmintic responses. For this reason, we compared all slopes in a pairwise fashion. Out of the 390 total strain-specific slope comparisons, we observed 275 pairwise instances across the 26 compounds where one strain had a significantly different slope than another strain (**S4 Table**). The variation in strain-specific slope comparisons further supports how the incorporation of genetic diversity is necessary to identify anthelmintic responses within a species. Here, we reinforce what

**Table 1. Drug class, subclass, and drugs used in this study.** Drug classes with defined Mode of Action (MoA) are listed.

| Drug Class | Drug Subclass | Drug | Target/ Mode of Action (MoA) |
|---|---|---|---|
| Benzimidazoles (BZ) | | Albendazole | β-Tubulin inhibitor |
| | | Benomyl | |
| | | Fenbendazole | |
| | | Mebendazole | |
| | | Thiabendazole | |
| Macrocyclic Lactones (ML) | Avermectins | Abamectin | Glutamate- and GABA-gated chloride channels receptor agonist |
| | | Doramectin | |
| | | Eprinomectin | |
| | | Ivermectin | |
| | | Selamectin | |
| | Milbemycins | Milbemycin oxime | |
| | | Moxidectin | |
| Nicotinic acetylcholine receptor (nAChR) agonists | Imidazothiazoles | Levamisole | Acetylcholine receptor agonist |
| | Tetrahydropyrimidines | Morantel citrate | |
| | | Pyrantel citrate | |
| Nicotinic acetylcholine receptor (nAChR) antagonists | Amino-acetonitrile Derivatives (AADs) | Monepantel sulfone LY33414916 | Acetylcholine receptor antagonist |
| | | Monepantel sulfide LY3348298 | |
| | Spiroindoles | Derquantel | |
| Crystal protein | | Cry5B | |
| Cyclicoctadepsipeptides | | Emodepside | |
| Schistosomicides | 1,3-thiazoles | Niridazole | |
| | Quinolines | Oxamniquine | |
| | Praziquantel | Praziquantel | |
| Other | Diethylcarbamazine | Diethylcarbamazine citrate | |
| | Pyrazinoisoquinolines | Piperazine | |
| | Salicylanilides | Closantel | |

previous studies have shown, that *C. elegans* is a powerful model for assessing the impact of genetic differences on phenotypic variation [23].

## Variation in response to BZs is driven by genetic differences among naturally diverse strains

Although BZs are essential in human and veterinary medicine, resistance to this drug class is prominent and common in natural parasite populations [24,25]. Historically, the mechanisms of nematode resistance to BZs were thought to have been limited to variants in the drug target beta-tubulin [26–29]. However, genetic differences in beta-tubulin genes do not explain all intraspecific and interspecific variation in BZs efficacy [30] or in responses to different BZs derivatives [31,32]. Genome-wide association studies (GWAS) of responses to albendazole, a widely used BZ, found quantitative trait loci (QTL) that do not overlap with beta-tubulin genes, suggesting that additional genes are involved in albendazole resistance [17]. Additionally, previous work, which included genetically diverse strains of *C. elegans* and *Caenorhabditis briggsae*, a closely related selfing species, found that conserved and drug-specific loci contribute to the effects of BZs (albendazole, fenbendazole, mebendazole, and thiabendazole) [33]. Because of evidence that additional genes beyond beta-tubulin genes are involved in BZs resistance, we have yet to fully understand the mechanisms of BZs resistance.

We assessed how natural variation contributes to phenotypic responses across five clinically relevant BZs (albendazole, benomyl, fenbendazole, mebendazole, and thiabendazole) that are widely used in human and veterinary medicine. The panel of six genetically divergent *C. elegans* wild strains was exposed to increasing concentrations of the five BZs (**S5 Table** and **Fig 2**). The strain MY16 displayed resistance in all five BZ dose-response curves, where the $EC_{10}$ for MY16 was significantly higher than $EC_{10}$ estimates from all the other strains in every BZ (**Fig 2**). The MY16 strain has a non-synonymous variant in the beta-tubulin gene *ben-1*, causing an amino acid change (A185P) [34] and a presumptive reduction in *ben-1* function. The other five strains do not have variants known to reduce *ben-1* function.

Benzimidazole strain-specific slope (*b*) estimates for each dose-response model varied but followed similar trends compared to $EC_{10}$ estimates (**Fig 3A and 3B**). These results suggest that the genetic differences among *C. elegans* strains mediate differential susceptibility across BZs. To quantify the degree of phenotypic variation attributable to segregating genetic differences among strains, we estimated broad-sense heritability ($H^2$) and narrow-sense heritability ($h^2$) of the phenotypic response for each dose of every BZ (see *Methods*; *Broad-sense and narrow-sense heritability calculations*) (**Fig 3C**). For example, we observed that $H^2$ ranged from 0 in 1 μM albendazole to 0.87 in 51.54 μM albendazole, and $h^2$ ranged from 0 in 1 μM albendazole to 0.73 in 51.54 μM albendazole. This heritable response indicated that genetic differences

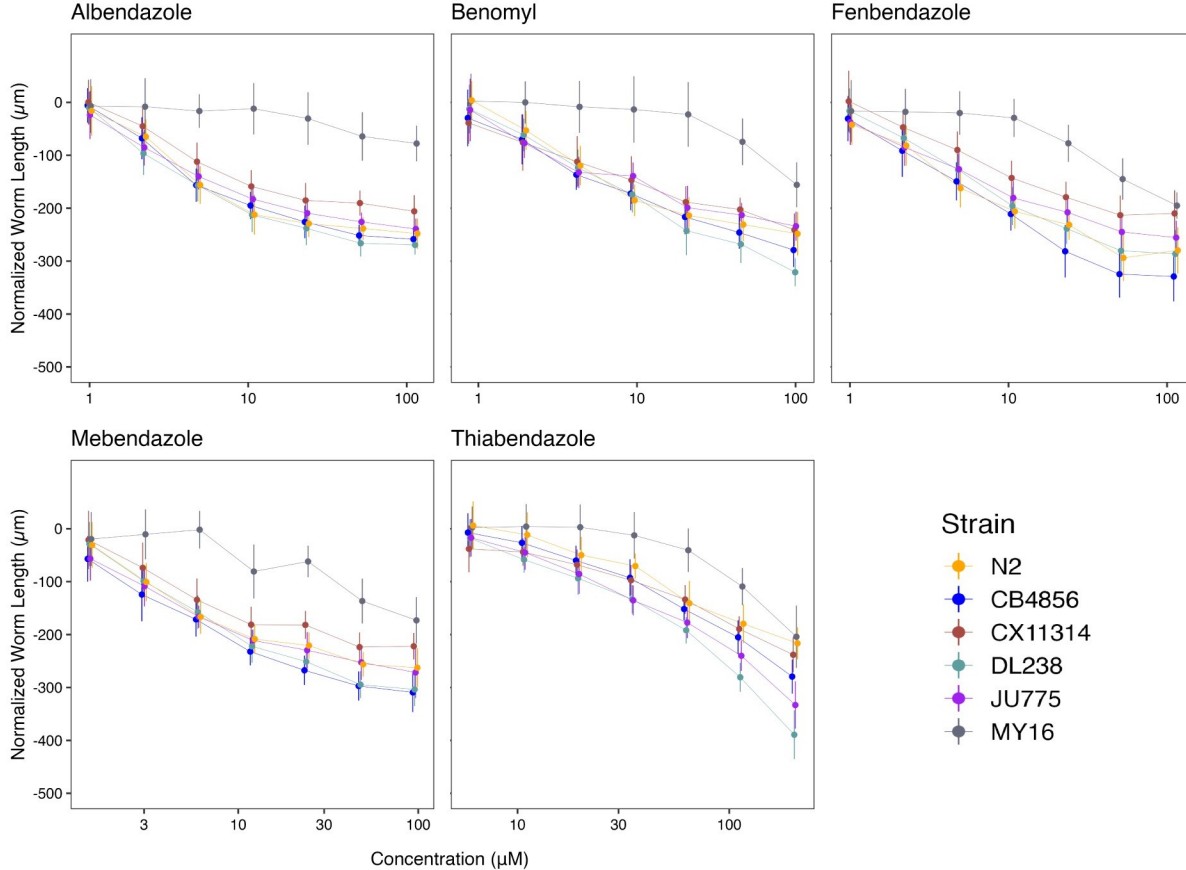

**Fig 2. Dose-response curves for benzimidazoles (BZs).** Normalized animal lengths (y-axis) are plotted for each strain as a function of the dose of benzimidazole supplied in the high-throughput phenotyping assay (x-axis); Albendazole, Benomyl, Fenbendazole, Mebendazole, and Thiabendazole. Strains are denoted by color. Lines extending vertically from points represent the standard deviation from the mean response. Statistical normalization of animal lengths is described in *Methods*.

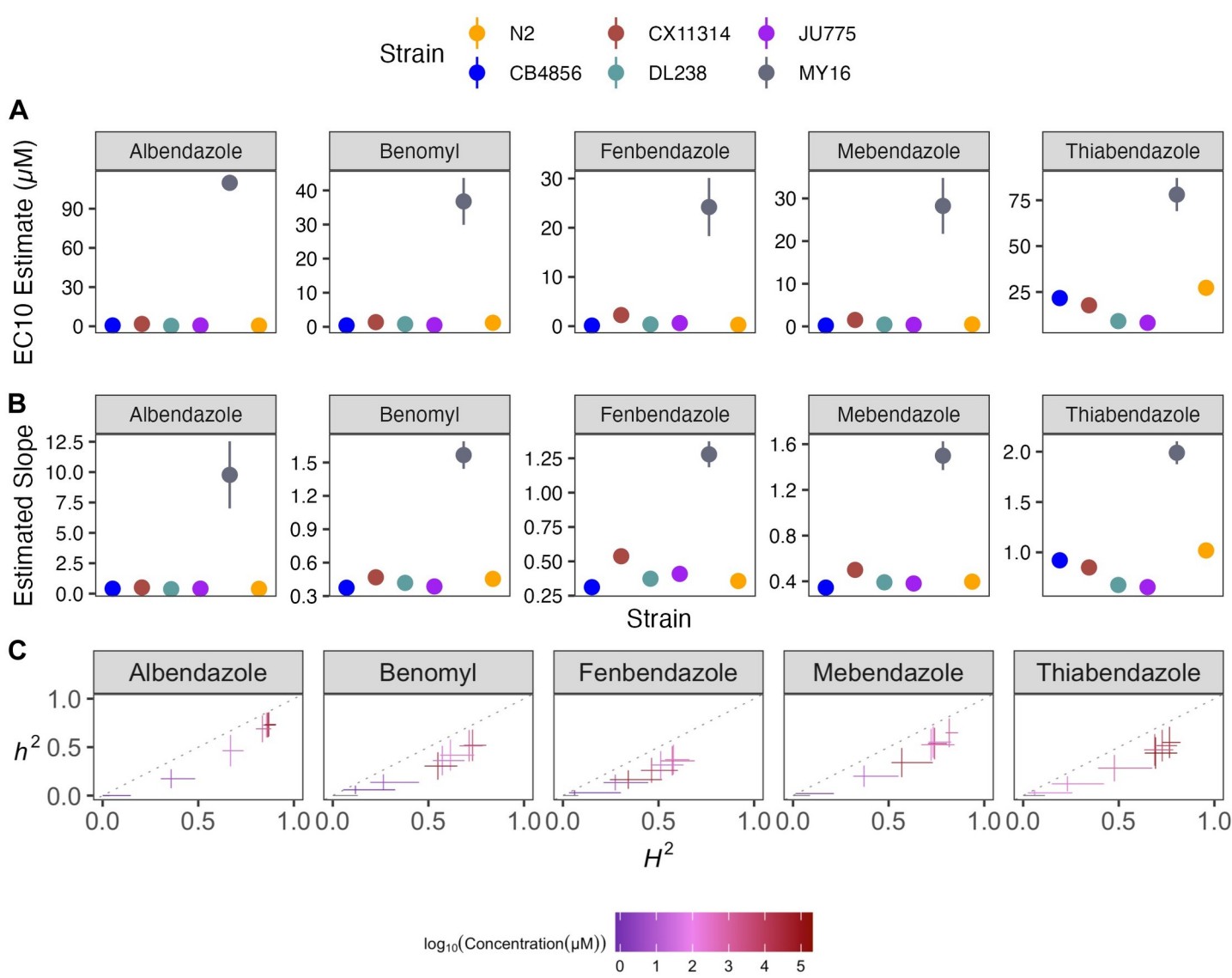

**Fig 3. Variation in benzimidazole (BZ) EC$_{10}$ dose-response and slope estimates can be explained by genetic variation across strains. (A)** Strain-specific EC$_{10}$ estimates (*e*) for each benzimidazole are displayed for each strain. Standard errors for each strain- and anthelmintic-specific EC$_{10}$ estimates are shown. **(B)** Strain-specific slope estimates (*b*) for each benzimidazole are displayed for each strain. Standard errors for each strain- and anthelmintic-specific slope estimate are indicated by the line extending vertically from each point. **(C)** The broad-sense (x-axis) and narrow-sense heritability (y-axis) of normalized animal length measurements were calculated for each concentration of each benzimidazole (*Methods*; *Broad-sense and narrow-sense heritability calculations*). The color of each cross corresponds to the log-transformed dose for which those calculations were performed. The horizontal line of the cross corresponds to the confidence interval of the broad-sense heritability estimate obtained by bootstrapping, and the vertical line of the cross corresponds to the standard error of the narrow-sense heritability estimate.

among the six strains underlie the variation in albendazole responses. Importantly, all five BZs had highly heritable responses, which indicates that the genetic diversity of *C. elegans* can be used to identify additional molecular mechanisms of BZs resistance beyond the *ben-1* beta-tubulin gene.

Because *ben-1* is not the only gene involved in BZs responses [17,33,35], we removed the MY16 strain from our analyses to observe how smaller genetic effects play a role in BZs responses in the other five strains (**S2 Fig**). After removing MY16, we observed that the strain CX11314 displayed the greatest resistance among the remaining five strains in all BZs except

thiabendazole. The strain N2 displayed the greatest resistance in thiabendazole after removing MY16. Also, the strains CB4856 and JU775, previously described as sensitive to BZs [18,34], displayed sensitivity across BZs and significant variability in thiabendazole, where the JU775 strain was more sensitive than the CB4856 strain (**S2 Fig**). Even after removing MY16, we found that responses for thiabendazole were highly heritable, although moderately heritable responses were observed for albendazole (**S2C Fig**). Benomyl, fenbendazole, and mebendazole had reduced heritability. These results support the previous findings that *ben-1* is not the only gene involved in BZs resistance and that diverse *C. elegans* strains vary across a spectrum of BZ responses [17,33,35]. The strain MY16 is a striking example of how natural BZ resistance alleles can protect nematodes from BZ treatment. In the context of natural parasitic nematode populations, it is easy to imagine how such beneficial alleles could spread rapidly and further exacerbate parasitic burdens. This example shows how genetic variation in a natural isolate can aid in the elucidation of resistance mechanisms that may not be found in N2 alone. It is important to note that Sydney Brenner could have been sent a different *C. elegans* isolate (*e.g.*, MY16) that is naturally resistant to BZs, and our understanding of BZ resistance would have been significantly delayed.

## Small variations in MLs dosage can significantly alter drug effectiveness among naturally diverse strains

The MLs comprise avermectins and milbemycins and are an essential class of anthelmintics because of our high dependence on them to control nematode parasites in livestock, companion animals, and humans [36]. Previous genetic screens performed in the *C. elegans* laboratory-adapted reference strain, N2, identified three genes that encode glutamate-gated chloride (GluCl) channel subunits (*glc-1*, *avr-14*, and *avr-15*) that are targeted by MLs [37,38]. Studies of abamectin have found additional loci involved in resistance [18]. By contrast, ML resistant parasitic nematode isolates do not have mutations in genes that encode GluCl channel subunits, suggesting additional mechanisms of resistance to MLs exist [39,40]. Quantitative genetic mappings in free-living and parasitic nematode species have identified genomic regions that confer drug resistance [17,18,33,40–43]. Altogether, mutations in GluCl channel genes have modest effects on some ML responses and do not explain the full spectrum of AR in this class. Other genes must be investigated to understand ML mechanisms of resistance.

Here, we assessed how natural variation contributes to phenotypic responses across seven MLs composed of five avermectins (abamectin, doramectin, eprinomectin, ivermectin, and selamectin) and two milbemycins (milbemycin and moxidectin) (**S5 Table** and **Fig 4**). We observed different susceptibility trends within and across avermectins and milbemycins. We found that the rank order among strains displaying the highest $EC_{10}$ varied among MLs. The strain DL238 had the highest $EC_{10}$ in eprinomectin and milbemycin. The strains CB4856 and N2 displayed the highest $EC_{10}$ in doramectin. The N2 strain displayed the highest $EC_{10}$ in selamectin (**Fig 5**). Ivermectin did not have significantly different $EC_{10}$ results among the six strains, suggesting that natural variation in these strains does not affect ivermectin resistance. Moxidectin had undefined $EC_{10}$ (estimates greater than the maximum exposure) and slope estimates, suggesting higher doses are needed to measure phenotypic responses across strains. Taken together, these results suggest that the genetic differences among *C. elegans* strains mediate differential susceptibilities across the majority of MLs.

To quantify the degree of phenotypic variation attributable to segregating genetic differences among strains, we estimated the $H^2$ and $h^2$ of the phenotypic responses in all MLs (**Fig 5C**). We observed that $H^2$ ranged from 0.02 in 0.00533 μM ivermectin to 0.87 in 0.27 μM milbemycin, and $h^2$ ranged from 0.01 in 0.00105 μM doramectin to 0.73 in 0.27 μM milbemycin.

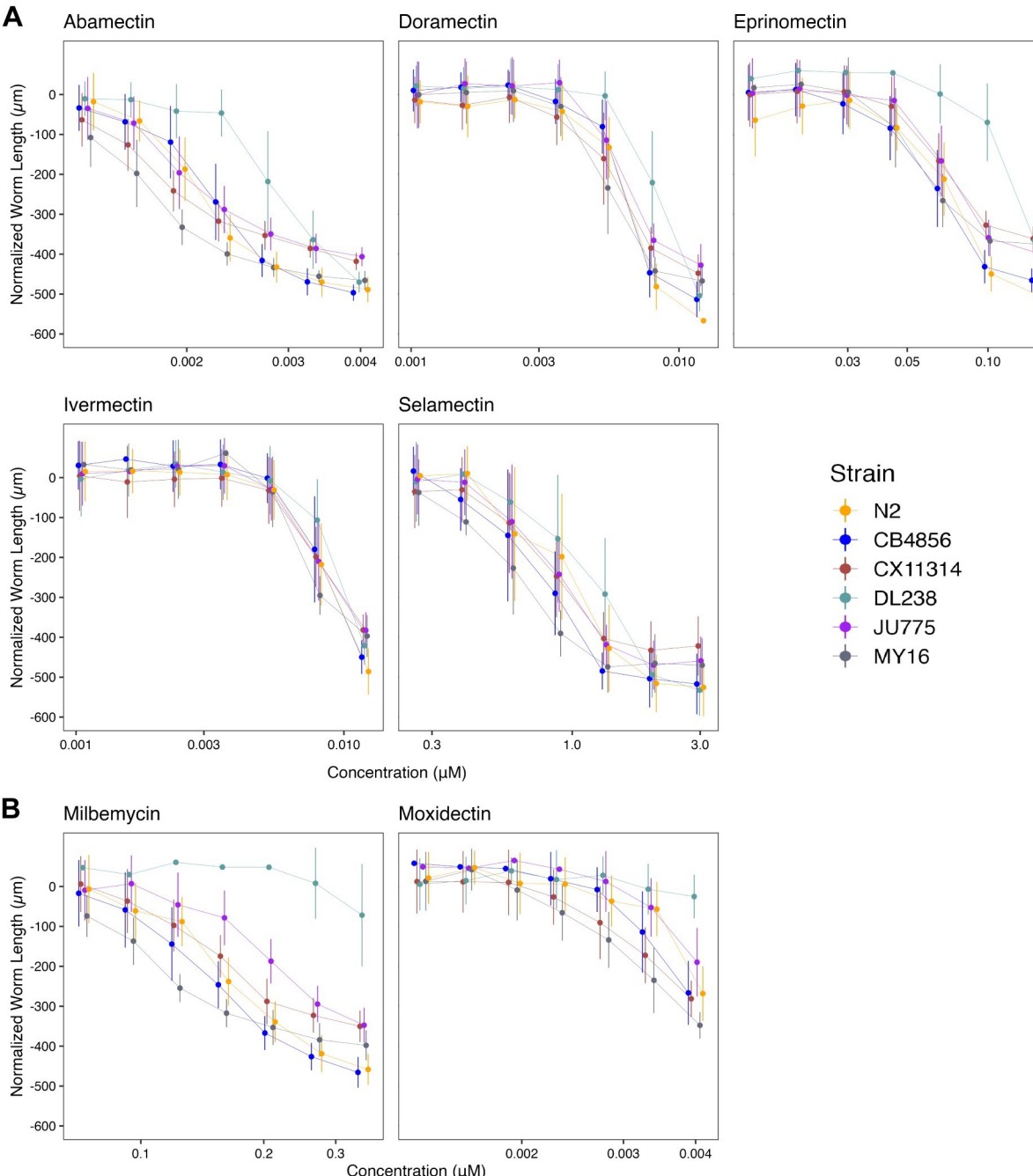

**Fig 4. Dose-response curves for macrocyclic lactones (MLs).** Normalized animal lengths (y-axis) are plotted for each strain as a function of the dose of macrocyclic lactone supplied in the high-throughput phenotyping assay (x-axis). Macrocyclic lactones are organized by **(A)** Avermectins: Abamectin, Doramectin, Eprinomectin, Ivermectin, Selamectin; and **(B)** Milbemycins: Milbemycin and Moxidectin. Strains are denoted by color. Lines extending vertically from points represent the standard deviation from the mean response. Statistical normalization of animal lengths is described in *Methods*.

Heritability for moxidectin could not be calculated because modeling produced undefined EC$_{10}$ and slope estimates (**Fig 4B**). In this strain set, we found milbemycin had the highest heritability estimate, whereas ivermectin and selamectin had the lowest heritability estimates of

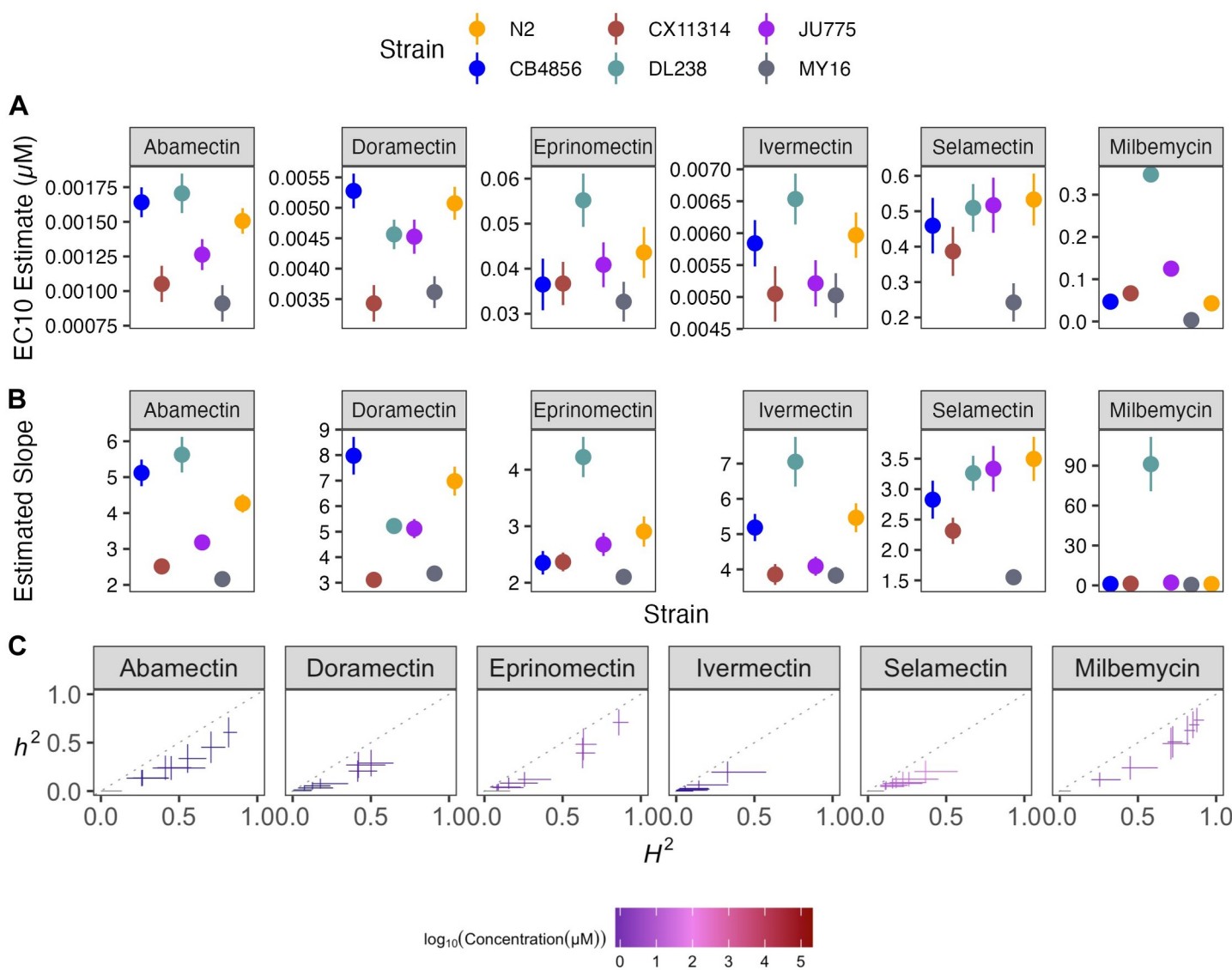

**Fig 5. Variation in macrocyclic lactone (ML) EC$_{10}$ dose-response and slope estimates can be explained by genetic variation across strains. (A)** Strain-specific EC$_{10}$ estimates ($e$) for each macrocyclic lactone are displayed for each strain. Standard errors for each strain- and anthelmintic-specific EC$_{10}$ estimates are indicated by the line extending vertically from each point. **(B)** Strain-specific slope estimates ($b$) for each macrocyclic lactone are displayed for each strain. Standard errors for each strain- and anthelmintic-specific slope estimate are indicated by the line extending vertically from each point. **(C)** The broad-sense (x-axis) and narrow-sense heritability (y-axis) of normalized animal length measurements were calculated for each concentration of each macrocyclic lactone (*Methods*; *Broad-sense and narrow-sense heritability calculations*). The color of each cross corresponds to the log-transformed dose for which those calculations were performed. The horizontal line of the cross corresponds to the confidence interval of the broad-sense heritability estimate obtained by bootstrapping, and the vertical line of the cross corresponds to the standard error of the narrow-sense heritability estimate. EC$_{10}$, estimated slope, and heritability could not be calculated for moxidectin and therefore, not plotted.

the MLs, indicating genetic variants between the six strains are involved in milbemycin response.

Another important factor in AR is suboptimal dosing of anthelmintics. Mis-dosing can cause variability in how the drug reaches targeted nematodes and causes an insufficient anthelmintic dose, which allows parasitic nematode populations to develop AR [44]. Although error-prone dosing methods can impact AR across all drug classes, it may be particularly important for MLs because small changes can cause vastly different anthelmintic responses. Here, we showed that small changes in MLs doses can significantly vary in effectiveness because of the

steep response curves (**Fig 4**). Additionally, the bioavailability of an anthelmintic and the length of exposure time also play a role in the dosage required to eliminate parasitic nematodes [45]. Correct dosing and appropriate bioavailability are critical in all anthelmintic treatments, but this point is even more striking in the MLs where the effective dose range is small.

## Natural genetic variation across *C. elegans* strains explains nAChR agonists responses

The nematode nAChRs in muscle cells are the targets of the cholinergic agonists (*e.g.*, levamisole, pyrantel, and morantel [46]). These nAChR agonists cause ligand-gated ion channels to open, producing prolonged muscle contraction and spastic paralysis in nematodes [46]. Levamisole-sensitive nAChR subunits have been identified in the parasite *Ascaris suum* where three distinct pharmacological nAChR subtypes are present on muscle cells [47,48] and in *C. elegans* where mutations in the nAChR subunit genes *unc-29*, *unc-38*, *unc-63*, *lev-1*, and *lev-8* affect sensitivity [49–52].

Here, we assessed how natural variation contributes to phenotypic responses across three nAChR agonists composed of tetrahydropyrimidines (morantel and pyrantel) and an imidazothiazole (levamisole) (**S5 Table** and **Fig 6**). The nAChR agonists strain-specific slope (*b*)

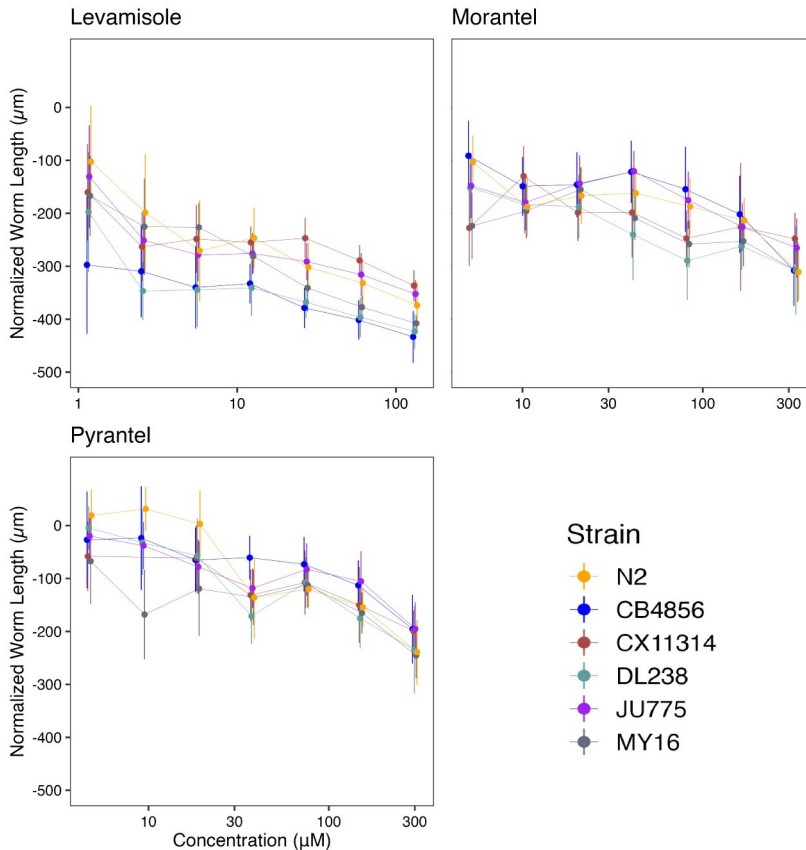

**Fig 6. Dose-response curves for nicotinic acetylcholine receptor (nAChR) agonists.** Normalized animal lengths (y-axis) are plotted for each strain as a function of the dose of anthelmintic supplied in the high-throughput phenotyping assay (x-axis); Levamisole, Morantel, and Pyrantel. Strains are denoted by color. Lines extending vertically from points represent the standard deviation from the mean response. Statistical normalization of animal lengths is described in *Methods*.

estimates for each dose-response model varied but followed near identical trends compared to $EC_{10}$ estimates, indicating genetic variation is responsible for the observed response variation (**Fig 7**). The strain DL238 had the highest $EC_{10}$ in levamisole. We found that the strain CB4856 had the highest $EC_{10}$ for both tetrahydropyrimidines, whereas the CX11314 and MY16 strains had the lowest $EC_{10}$ and are thus the most susceptible. Variable patterns across strains within the same drug class suggest nAChR agonists might be acting on different genetic targets.

$EC_{10}$ and strain-specific slope ($b$) estimates suggested that the genetic differences among *C. elegans* strains mediate differential susceptibility across nAChR agonists. To quantify the degree of phenotypic variation attributable to segregating genetic differences among strains, we estimated the $H^2$ and $h^2$ of the phenotypic response for each dose of every nAChR agonist (**Fig 7C**). We observed that $H^2$ ranged from 0.06 in 300 μM pyrantel to 0.52 in 282.5 μM morantel, and $h^2$ ranged from 0.028 in 300 μM pyrantel to 0.34 in 82.5 μM morantel. We found morantel to elicit the most heritable response, whereas pyrantel elicited the lowest heritable response of the nAChR agonists. Variable heritability in the tetrahydropyrimidines indicates that nAChR agonists may be acting on different genetic targets. Even if drugs have similar trends in $EC_{10}$ and slope, heritability might help identify drugs where phenotypic variance in response to anthelmintic treatment is attributable to genetic differences. Additionally, although we have several genetic targets identified in *C. elegans*, it is unclear whether nAChR gene families remain highly conserved among nematode species or whether different species-specific functions can be exploited as potential targets for the control of particular parasites [53].

## Dose-response assessments can be used across genetically diverse strains to identify anticipated anthelmintic effectiveness in combination therapies

Drugs outside of the three major anthelmintic classes are valuable because they have different hypothesized targets and mechanisms of resistance that could be effective against nematodes resistant to other drug classes. Drugs with different mechanisms of resistance can be used in combination therapies with other anthelmintics to create a more effective treatment. Although each anthelmintic class has different molecular targets, it is not well understood how a strain resistant to one class responds to another class. Here, in addition to the three major anthelmintic classes, we exposed strains to five different groups of anthelmintics categorized by their hypothesized drug targets. Nematode phenotypic responses were measured against nAChR antagonists (monepantel sulfone, monepantel sulfide, and derquantel), a pore-forming crystal toxin (Cry5B), a cyclicoctadepsipeptide (emodepside), schistosomicides (niridazole, oxamniquine, and praziquantel), a salicylanilide (closantel), diethylcarbamazine, and piperazine (**S3–S13** Figs). By assessing nematode response patterns to individual drugs, we can identify which drugs could be paired in combination therapies.

In the past few decades, Cry5B and the nAChR agonist (Levamisole) have been used in combination therapy as strains resistant to nAChR agonists were susceptible to Cry5B [54]. Here, we find that the CB4856, DL238, and MY16 strains were sensitive to Cry5B, whereas the CX11314 and JU775 strains were sensitive to levamisole (**Figs 8 and S14 and S15**). We observed different patterns of susceptibility (strain rank order) between levamisole and Cry5B, indicating that this combined therapy could be an effective drug combination. Another promising combination therapy is derquantel and abamectin [55,56]. Derquantel and abamectin have been used in combination to treat multi-drug resistant *Haemonchus contortus* [57,58]. However, studies have found monepantel to be more effective than the combined derquantel and abamectin treatment, although monepantel resistance is also prevalent, further exaggerating resistance issues in *H. contortus* [55,59,60]. Here, we observed that the strain MY16 was most sensitive to abamectin, whereas the CB4856 and DL238 strains were most resistant. In

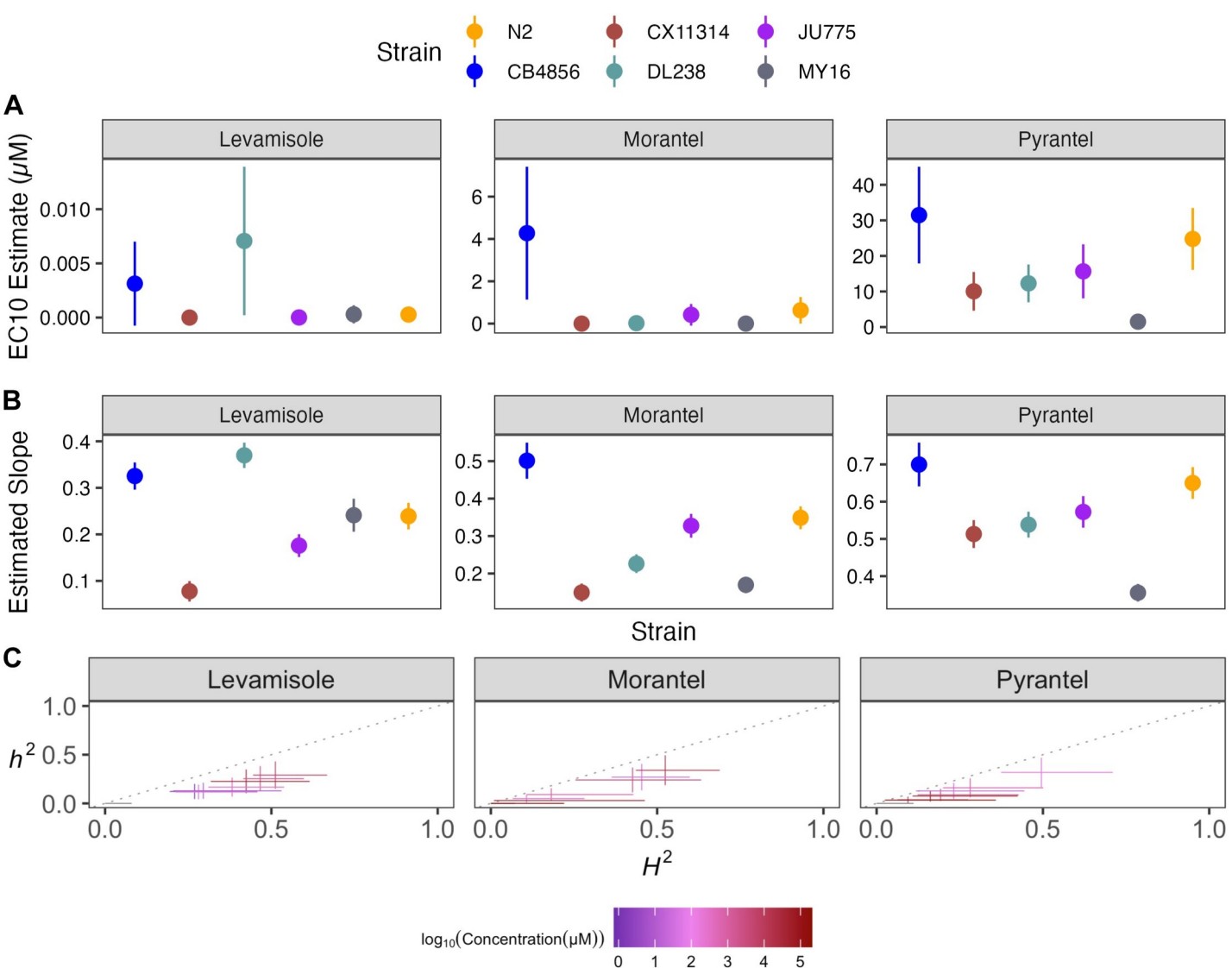

**Fig 7. Variation in nicotinic acetylcholine receptor agonists (nAChR agonists) EC$_{10}$ dose-response and slope estimates can be explained by genetic variation across strains. (A)** Strain-specific EC$_{10}$ estimates (*e*) for each nicotinic acetylcholine receptor agonist are displayed for each strain. Standard errors for each strain- and anthelmintic-specific EC$_{10}$ estimates are indicated by the line extending vertically from each point. **(B)** Strain-specific slope estimates (*b*) for each nicotinic acetylcholine receptor agonist are displayed for each strain. Standard errors for each strain- and anthelmintic-specific slope estimate are indicated by the line extending vertically from each point. **(C)** The broad-sense (x-axis) and narrow-sense heritability (y-axis) of normalized animal length measurements were calculated for each concentration of each nicotinic acetylcholine receptor agonist (*Methods*; *Broad-sense and narrow-sense heritability calculations*). The color of each cross corresponds to the log-transformed dose for which those calculations were performed. The horizontal line of the cross corresponds to the confidence interval of the broad-sense heritability estimate obtained by bootstrapping, and the vertical line of the cross corresponds to the standard error of the narrow-sense heritability estimates.

derquantel, we found that the strains JU775 and MY16 were the most sensitive. Comparatively, the strain JU775 showed significant sensitivity to the monepantel drugs (monepantel sulfone, monepantel sulfide) (**S8 and S9 Figs**). Here, patterns of susceptibility and resistance indicated that combination therapy composed of derquantel and abamectin would be a more effective treatment than monepantel alone. Lastly, emodepside has also been commercialized and approved for anthelmintic treatments in companion animals in combination with praziquantel [61]. Here, we find that emodepside had heritable responses (**S7 and S16 Figs**) among genetically diverse *C. elegans*, but praziquantel had no heritable responses (**S13 Fig**). Although

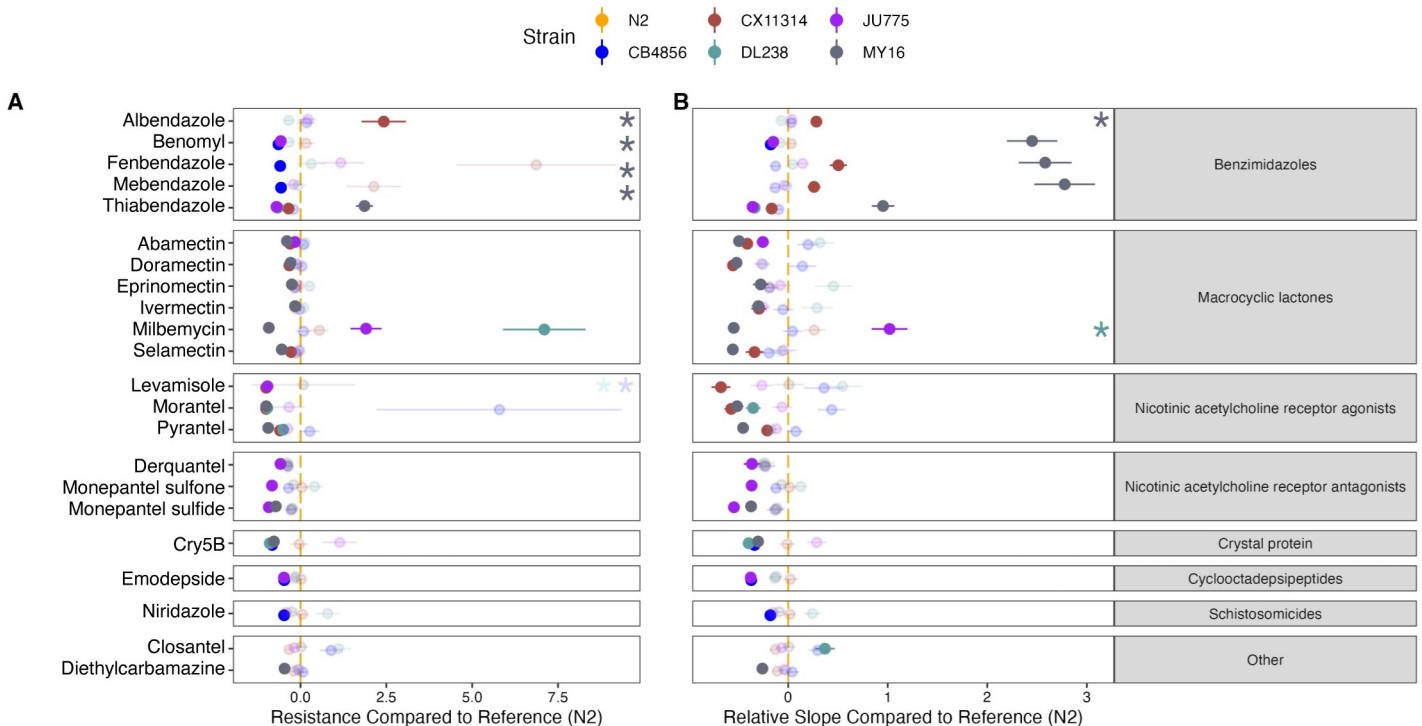

**Fig 8. Variation in EC$_{10}$ and dose-response slope estimates can be explained by genetic variation across strains. (A)** The relative potency of each anthelmintic for each strain compared to the N2 strain is shown. Solid points denote strains with significantly different relative resistance to that anthelmintic compared to the N2 strain (Student's t-test and subsequent Bonferroni correction with a $p_{adj} < 0.05$). Faded points denote strains not significantly different than the N2 strain. Asterisks denote strains with normalized estimates greater than ten compared to the N2 strain. See **S14 Fig** for the relative potency of all strains in each anthelmintic. Anthelmintic drugs with undefined EC$_{10}$ estimates (estimates greater than the maximum dose to which animals were exposed) are not shown. **(B)** For each anthelmintic, the relative steepness of the dose-response slope inferred for that strain compared to the N2 strain is shown. Solid points denote strains with significantly different dose-response slopes for that anthelmintic compared to the N2 strain (Student's t-test and subsequent Bonferroni correction with a $p_{adj} < 0.05$). Faded points denote strains without significantly different slopes than the N2 strain. Asterisks represent strains with slope estimates greater than 20 compared to the N2 strain. See **S15 Fig** for the slope estimates of all strains in each anthelmintic. Anthelmintic drugs with undefined slope estimates are not shown. The broad class to which each anthelmintic belongs is denoted by the strip label for each facet.

we describe anticipated anthelmintic effectiveness in combination therapies, we acknowledge that a limitation of this study is the small number of strains used. Even though we used a genetically divergent strain set, we have captured a fraction of the genetic variation across the *C. elegans* species. Promising combination therapies can be tested using larger strain sets. Altogether, dose-response assessments in *C. elegans* provide a useful platform to assess hypothesized effectiveness of drugs that can be used together in combination therapies.

### Dose-response assessment across genetically diverse stains identifies five anthelmintics for which *C. elegans* had little to no phenotypic responses

Because *C. elegans* is an inexpensive and highly tractable model, we could quickly assess which drugs we should continue to study in this model and drugs that will likely not provide useful results. Dose-response curves for the schistosomicides (niridazole, oxamniquine, and praziquantel) showed little to no responses across *C. elegans* strains (**S10**, **S11** and **S13** **Figs**). Minimal responses were also observed for diethylcarbamazine and piperazine (**S6** and **S12** **Figs**). With minimal to no response for oxamniquine, praziquantel, and piperazine, EC$_{10}$ and slope estimates could not be calculated. The schistosomicides have previously been shown to have no activity against nematodes [62], and thus it was not surprising that we observed little

response in *C. elegans*. Diethylcarbamazine contains a piperazine ring that is essential for the activity of the drug and is the treatment of choice for lymphatic filariasis and loiasis [63]. Because piperazine did not cause a response in *C. elegans*, it is not surprising that diethylcarbamazine did not as well. Although little to no responses (*i.e.*, nematode development defects) were observed for five drugs using our assays, it is possible assays measuring different fitness traits could elicit anthelmintic effects. Overall, *C. elegans* is a useful model for screening potential nematicides.

## *C. elegans* is an invaluable model for understanding anthelmintic drug target identification and characterization

Our assays measured *C. elegans* developmental delay in the presence of anthelmintics across multiple concentrations of each drug. This study yielded several major findings. First, dose-response experiments captured the variation in development across six diverse strains within the *C. elegans* species. Dose-response trends yielded information that can be used to assess how other nematodes might respond to the tested anthelmintic drugs. Second, the large-scale HTA provided quantitative data with the required statistical power and sample sizes needed to effectively measure anthelmintic responses. Third, we were able to identify which drugs had heritable responses and, of those drugs, which doses were most heritable. The most heritable doses of each drug can be used in downstream GWAS to identify genomic regions correlated with AR [4]. By narrowing genomic intervals, subsequent candidate genes can be identified, edited, and validated for anthelmintic responses to ultimately identify the genetic variants involved in resistance and inform downstream parasitic nematode treatments [18,33].

It is well established that parasitic nematodes are more genetically diverse than *C. elegans* and infect virtually all animal species, so understanding the role genetic diversity plays in anthelmintic resistance is critical [7]. The flow of resistance alleles within and among parasite populations has profound implications for the epidemiology of host infection, disease presentation, and the responses of parasite populations to selection pressures, such as anthelmintic treatment [64,65]. Additionally, the development of novel anthelmintics is slow, expensive, and complex, making it critical to correctly apply and monitor the usage of our existing drugs. We suggest that genetically diverse *C. elegans* strains should be deployed to aid high-throughput anthelmintic screening efforts to identify effective anthelmintics and estimated effective concentrations to use when testing in parasitic nematodes.

The presented data focused on the natural genetic variation in *C. elegans* and will require additional studies to identify genes responsible for the observed anthelmintic responses. This study summarized anthelmintic responses in naturally diverse *C. elegans* strains and highlighted drugs to focus on in downstream studies. Ultimately, AR gene variants responsible for observed effects need to be identified and validated in *C. elegans* and subsequently tested in parasitic nematodes.

## Materials and methods

### *C. elegans* strain selection and maintenance

Six *Caenorhabditis elegans* strains (CB4856, CX11314, DL238, JU775, MY16, PD1074) from the *C. elegans* Natural Diversity Resource (CeNDR) were used in this study [14]. Isolation details for the six strains are included in CeNDR. These strains were selected from the CeNDR divergent strain set, where the strain PD1074 is referred to by its isotype name, N2. In CeNDR, strains that are >99.97% genetically identical are grouped into isotypes, PD1074 and N2 are nearly genetically identical, therefore, we chose to label PD1074 as N2 to illustrate the response

of the canonical laboratory strain N2 [66]. These six strains represent 74% of the variation across the *C. elegans* species. Before measuring anthelmintic responses, animals were maintained at 20˚C on 6 cm plates with modified nematode growth medium (NGMA), which contains 1% agar and 0.7% agarose to prevent animals from burrowing. The NGMA plates were seeded with the *Escherichia coli* strain OP50 as a nematode food source. All strains were grown for three generations without starvation on the NGMA plates before anthelmintic exposure to reduce the transgenerational effects of starvation stress. The specific growth conditions for nematodes used in the high-throughput anthelmintic response assays are described below.

## Nematode food preparation

Detailed nematode food preparation steps were followed as previously described [23]. One batch of HB101 *E. coli* was used as a nematode food source for all assays. Briefly, a frozen stock of HB101 *E. coli* was used to inoculate and grow a one-liter culture at an $OD_{600}$ value of 0.001. A total of 14 cultures containing one liter of pre-warmed 1x Horvitz Super Broth (HSB) and an $OD_{600}$ inoculum grew for 15 hours at 37˚C until cultures were in the late log growth phase. After 15 hours, flasks were removed from the incubator and transferred to 4˚C to arrest growth. Cultures went through three rounds of centrifugation, where the supernatant was removed, and the bacterial cells were pelleted. Bacterial cells were washed and resuspended in K medium. The $OD_{600}$ value of the bacterial suspension was measured and diluted to a final concentration of $OD_{600}100$ with K medium, aliquoted to 15 ml conical tubes, and stored at -80˚C for use in the anthelmintic dose-response assays.

## Anthelmintic stock preparation

All 26 anthelmintic stock solutions were prepared using either dimethyl sulfoxide (DMSO) or water, depending on the anthelmintic's solubility. Sources, catalog numbers, stock concentrations, and preparation for each anthelmintic are provided (**S5 Table**). Anthelmintic stock solutions were prepared, aliquoted, and stored at -20˚C for use in the dose-response assays.

## High-throughput anthelmintic dose-response assay

For each assay, populations of each strain were amplified and bleach-synchronized in triplicate. The bleach synchronization was replicated to control for variation in embryo survival and subsequent effects on developmental rates that could be attributed to bleach effects. After bleach synchronization, approximately 30 embryos were dispensed into the wells of a 96-well microplate in 50 μL of K medium. Strains were randomly assigned to columns of the 96-well microplates to vary strain column assignments across the replicate bleaches. Each strain was present in duplicate on each plate. Four replicate 96-well microplates within each of the three bleach replicates for each anthelmintic and control condition tested in the assay were prepared, labeled, and sealed using gas-permeable sealing films (Fisher Cat # 14-222-043). Plates were placed in humidity chambers to incubate overnight at 20˚C while shaking at 170 rpm (INFORS HT Multitron shaker). The following morning, food was prepared to feed the developmentally arrested first larval stage animals (L1s) using the required number of $OD_{600}100$ HB101 aliquots (see *Nematode food preparation*). The aliquots were thawed at room temperature, combined into a single conical tube, and diluted to an $OD_{600}30$ with K medium. To inhibit further bacterial growth and prevent contamination, 150 μM of Kanamycin was added to the HB101. Working with a single anthelmintic at a time, an aliquot of anthelmintic stock solution thawed at room temperature (see *Anthelmintic stock preparation*) and was diluted to a working concentration. The anthelmintic working concentration was set to the concentration that would give the highest desired dose when added to the 96-well microplates at 1% of the

total well volume. The serial dilution of the anthelmintic working solution was prepared using the same diluent, DMSO or water, used to make the stock solution. The dilution factors ranged from 1.2 to 2.5 depending on the anthelmintic used, but all serial dilutions had eight concentrations, including a 0 μM control (**S5 Table**). The serial dilution was then added to an aliquot of the $OD_{600}$30 K medium at a 3% volume/volume ratio. Next, 25 μl of the food and anthelmintic mixture was transferred into wells of the 96-well microplates to feed the arrested L1s at a final HB101 concentration of $OD_{600}$10 and expose L1 larvae to an anthelmintic at one of eight levels of the dilution series. Immediately afterward, the 96-well microplates were sealed using a new gas permeable sealing film, returned to the humidity chambers, and incubated for 48 hours at 20˚C shaking at 170 rpm. The remaining 96-well microplates were fed and exposed to anthelmintics in the same manner. After 48 hours of incubation in the presence of food and anthelmintic, the 96-well microplates were removed from the incubator and treated with 50 mM sodium azide in M9 for 10 minutes to paralyze and straighten nematodes. Images of nematodes in the microplates were immediately captured using a Molecular Devices ImageXpress Nano microscope (Molecular Devices, San Jose, CA) using a 2X objective. The ImageXpress Nano microscope acquires brightfield images using a 4.7 megapixel CMOS camera and stores images in a 16-bit TIFF format. The images were used to quantify the development of nematodes in the presence of anthelmintics as described below (see *Data collection* and *Data cleaning*).

## Data processing

CellProfiler software (Version 4.0.3) was used to quantify animal lengths from images collected on the Molecular Devices ImageXpress Nano microscope [67]. A Nextflow pipeline (Version 20.01.0) was written to run command-line instances of CellProfiler in parallel on the Quest High-Performance Computing Cluster (Northwestern University). The CellProfiler workflow can be found at (https://github.com/AndersenLab/cellprofiler-nf). CellProfiler modules and Worm Toolbox were developed to extract morphological features of individual *C. elegans* animals from images from the HTA [68]. The custom CellProfiler pipeline generates animal measurements by using four worm models: three worm models tailored to capture animals at the L4 larval stage, in the L2 and L3 larval stages, and the L1 larval stage, respectively, as well as a "multi-drug high dose" (MDHD) model, to capture animals with more abnormal body sizes caused by extreme anthelmintic responses. Nematodes grown under control conditions or those unperturbed by an anthelmintic developed at a normal rate to the L4 larval stage. Nematodes affected by anthelmintics have delayed development. Worm model estimates and custom CellProfiler pipelines were written using the WormToolbox in the GUI-based instance of CellProfiler [69]. Next, a custom R package, *easyXpress* (Version 1.0), was then used to process animal measurements output from CellProfiler [15]. These measurements comprised our raw dataset.

## Data cleaning

The presented analysis has been modified from previous work [23]. All analyses were performed using the R statistical environment (version 4.2.1) unless stated otherwise. The high-throughout anthelmintic dose-response assay produced thousands of images per experimental block; thus, we implemented a systematic approach to assess the quality of animal measurement data in each well. Several steps were implemented to clean the raw image data using metrics indicative of high-quality animal measurements for downstream analysis.

1. Objects with a *Worm_Length* > 30 pixels, 100 microns, were removed from the CellProfiler data to (A) retain L1 and MDHD-sized animals and (B) remove unwanted particles [70]. Using the *Worm_Length* > 30 pixels threshold to retain small sensitive animals, more small objects, such as debris (*i.e.*, bacterial clumps, contaminants, drug crystals), were also retained (**S17 Fig**).

2. *R/easyXpress* [15] was used to filter measurements from worm objects within individual wells with statistical outliers and to parse measurements from multiple worm models down to single measurements for single animals.

3. The data were visualized by drug, drug concentration, assay, strain, and worm model for two purposes. First, to ensure that each drug, by assay, contained control wells that had a *mean_wormlength_um* between 600–800 μm, the size of an L4 animal. If the *mean_wormlength_um* in the control wells was not between the 600–800 μm range, then that strain and/or assay were removed for the drug (**S18 Fig**). This filter ensured the control wells, DMSO or water, primarily contained L4 animals. Assays and drugs that did not meet the control well *mean_wormlength_um* criteria and were thus subsequently removed were: abamectin (assay A), derquantel (assay H), niridazole (assay H), eprinomectin (assay I), and piperazine (assay I). Second, we wanted to identify drugs that contained a high abundance of MDHD model objects across all assays and drug concentrations. Drugs with an abundance of objects classified by the MDHD model across assays and concentrations likely contain debris (*i.e.*, bacterial clumps, contaminants, or drug crystals). The MDHD model was removed from the following 13 drugs to limit debris and small objects: albendazole, benomyl, Cry5B, diethylcarbamazine, fenbendazole, mebendazole, morantel, niridazole, oxamniquine, piperazine, praziquantel, pyrantel, and thiabendazole.

4. We then filtered the data to wells containing between three and forty animals, under the null hypothesis that the number of animals is an approximation of the expected number of embryos originally titered into wells (approximately 30). Given that our analysis relied on well median animal length measurements, we excluded wells with fewer than three animals to reduce sampling error.

5. We removed statistical outlier measurements within each concentration for each strain for every anthelmintic drug to reduce the likelihood that statistical outliers influence anthelmintic dose-response curve fits.

6. Next, we removed measurements from all doses of each anthelmintic drug that were no longer represented in at least 80% of the independent assays because of previous data filtering steps.

7. Finally, we normalized the data by (1) regressing variation attributable to assay and technical replicate effects and (2) normalizing these extracted residual values to the average control phenotype. For each anthelmintic drug, we estimated a linear model using the raw phenotype measurement as the response variable, and both assay and technical replicate identity as explanatory variables following the formula *median_wormlength_um ~ Metadata_Experiment + bleach* using the *lm()* function in base R. We used the residuals from the linear model to remove the effect of assay and bleach from the raw phenotypes. Next, for each drug, we calculated the mean of residual values in control conditions for each strain in each assay and bleach. Finally, for each drug, strain, assay, and bleach, we subtracted the appropriate mean control values from the model residuals to arrive at our normalized length measurements, which were used in all downstream statistical analyses. These normalized length measurements have the helpful property of being centered on zero in

control conditions for each strain and, therefore, control for natural differences in the length of the strains.

## Small object removal

In previous analyses, we used *Worm_Length > 50* (165 microns) to filter out small objects from data before performing cleaning steps [23]. For the anthelmintics, we saw that when applying this filter, high dose concentrations for 12 of the 26 anthelmintics were filtered and removed. Additionally, the anthelmintic selamectin was entirely removed from the dataset (**S17 Fig**). Although a *Worm_Length > 50* filtered debris from image data, it also filtered small drug-affected nematodes, which were abundant in this study. To ensure that we captured small drug-affected nematodes across anthelmintics and minimized the amount of retained debris, we altered the animal length threshold to *Worm_Length > 30* (100 microns). The threshold *Worm_Length > 30* was previously recorded as the smallest animal length of L1 animals after an hour of feeding [70]. To confirm that we were retaining animal objects, we (1) retained the MDHD model for drugs that had small animals present at high dose concentrations (see *Methods* and *Data cleaning*) and (2) observed high dose well images to ensure the MDHD model was identifying nematodes.

## Dose-response model estimation and statistics

After *Data cleaning* and *Small object removal* steps, each anthelmintic dose contained a minimum of 10 replicates per strain with a minimum of 30 nematodes, to proceed with dose-response model estimates and statistics. Dose-response model estimates and statistics have been modified from previous work [23]. We estimated overall and strain-specific dose-response models for each anthelmintic by fitting a log-logistic regression model using *R/drc* (Version 3.0.1) [71]. The four-parameter log-logistic function, *LL.4*, fits the anthelmintic data best. The *LL.4* model was fit to each anthelmintic using the *drc::drm()* function, where the model specified the following parameters: $b$, the slope of the dose-response curve; $c$, the upper asymptote of the dose-response curve; $d$, the lower asymptote of the dose-response curve; and $e$, the effective dose [72]. Strain was specified as a covariate for parameters $b$ and $e$, allowing us to estimate strain-specific dose-response slopes and effective doses. The lower asymptote, $d$, was specified at -600, the theoretical normalized length of animals at the first larval stage.

The *drc::ED()* function was used to extract strain-specific $EC_{10}$ values and strain-specific slope values [71]. We quantified the relative susceptibilities of each strain pair for each compound based on their estimated $EC_{10}$ values using the *drc::EDcomp()* function, which used an approximate *F*-test to determine whether the variances (represented by delta-specified confidence intervals) calculated for each strain-specific dose-response model's $e$ parameter estimates were significantly different. We quantified the relative slope steepness of dose-response models estimated for each strain within each compound using the *drc::compParm()* function, which used a *z*-test to compare the means of each $b$ parameter estimate. Results shown were filtered to comparisons against N2 dose-response parameters (**Fig 8**). Significantly different estimates in both cases were determined by correcting to a family-wise type I error rate of 0.05 using a Bonferroni correction. To determine whether strains were significantly more resistant or susceptible to more anthelmintics or anthelmintic classes by chance, we conducted 1000 Fisher exact tests using the *fisher.test()* function with 2000 Monte Carlo simulations.

### Broad-sense and narrow-sense heritability calculations

Phenotypic variance can be split into two parts, genetic variance ($V_G$) and residual variance ($V_E$). To determine the genetic contributions to phenotypic variance, we measured broad-sense ($H^2$) and narrow-sense heritability ($h^2$). Broad-sense heritability ($H^2$) is the total fraction of phenotypic variance explained by genetic variation in a population [4]. Here, $H^2$ was calculated using the equation, $H^2 = V_G / (V_G + V_E)$, where we extracted the among strain variance ($V_G$) and residual variance ($V_E$). Broad-sense heritability was estimated ($H^2$) using in the R statistical environment (version 4.0.4) using the R package *lme4* (v1.1–27.1), to fit a linear mixed-effects model to the normalized median animal length data using strain as a random effect [23]. Genetic variance ($V_G$) can be partitioned into additive ($V_A$) and non-additive ($V_{NA}$) variance components. Additive genetic variance ($V_A$) is the amount of genetic variance explained by genotype variants that differ in a specific population. Narrow-sense heritability ($h^2$) is the fraction of phenotypic variation explained by additive genetic variation in a population [4]. Narrow-sense heritability ($h^2$) measures the additive genetic variance ($V_A$) over the total phenotypic variance, $h^2 = V_A / V_P$. To calculate $h^2$, we first generated a strain matrix using the strain *genomatrix profile* on NemaScan (https://github.com/AndersenLab/NemaScan), a genome-wide association (GWA) mapping and simulation pipeline [73]. NemaScan uses the GCTA software suite to calculate genomic relationship matrices [74,75] and used the hard-filtered variant call format (VCF) file (WI.20220216.hard-filter.vcf.gz) generated in the 20220216 CeNDR release (https://www.elegansvariation.org/data/release/latest). The VCF used includes high-quality variants from 550 *C. elegans* isotypes, genetically distinct strains, after quality control filters. We then calculated $h^2$ using the *sommer* (v4.1.3) R package by calculating the variance-covariance matrix ($M_A$) from the strain matrix using the *sommer::A.mat* function [23]. We estimated $V_A$ using the linear mixed-effects model function *sommer::mmer* using strain as a random effect and $M_A$ as the covariance matrix. We then estimated $h^2$ and its standard error using the *sommer::vpredict* function (**S16 Fig**). Lastly, a correlation between the $h^2$ of the exposure closest to the estimated $EC_{10}$ and the drug doses that exhibited the maximum $h^2$ for each anthelmintic with definitive $EC_{10}$ estimates was performed (**S1 Fig** and **S6 Table**).

## Supporting information

**S1 Table. Strain-specific $EC_{10}$ estimates for each anthelmintic (μM units).**
(CSV)

**S2 Table. Strain-specific slope estimates for each anthelmintic.**
(CSV)

**S3 Table. Relative potency estimates in pairwise comparisons of $EC_{10}$ estimates among all strains for each anthelmintic.**
(CSV)

**S4 Table. Relative potency estimates in pairwise comparisons of slope estimates among all strains for each anthelmintic.**
(CSV)

**S5 Table. Anthelmintic drug stock solution preparation details and dosages used.**
(CSV)

**S6 Table. Anthelmintic $EC_{10}$ estimates and correlated heritability calculations.**
(CSV)

**S1 Fig. EC$_{10}$ estimates from genetically diverse strains predict exposures with heritable responses.** The log-transformed exposure that elicited the most heritable response to each anthelmintic (y-axis) is plotted against the log-transformed exposure of that same anthelmintic nearest to the inferred EC$_{10}$ from the dose-response assessment. The exposure closest to the EC$_{10}$ across all anthelmintics exhibited significant explanatory power to determine the exposure that elicited heritable phenotypic variation.
(TIF)

**S2 Fig. Variation in benzimidazole (BZ) EC$_{10}$ dose-response and slope estimates without MY16 demonstrate how small genetic effects have variable responses across strains. (A)** Strain-specific EC$_{10}$ estimates (*e*) for each benzimidazole are displayed for each strain. Standard errors for each strain- and anthelmintic-specific EC$_{10}$ estimates are shown. **(B)** Strain-specific slope estimates (*b*) for each benzimidazole are displayed for each strain. Standard errors for each strain- and anthelmintic-specific slope estimate are indicated by the line extending vertically from each point. **(C)** The broad-sense (x-axis) and narrow-sense heritability (y-axis) of normalized animal length measurements were calculated for each concentration of each benzimidazole (*Methods*; *Broad-sense and narrow-sense heritability calculations*). The color of each cross corresponds to the log-transformed dose for which those calculations were performed. The horizontal line of the cross corresponds to the confidence interval of the broad-sense heritability estimate obtained by bootstrapping, and the vertical line of the cross corresponds to the standard error of the narrow-sense heritability estimate.
(TIF)

**S3 Fig. Dose-response curve for closantel.** Normalized animal lengths (y-axis) are plotted for each strain as a function of the dose of closantel in the high-throughput microscopy assay (x-axis). Strains are denoted by color. Lines extending from points represent the standard deviations from the mean responses. Statistical normalization of animal lengths is described in *Methods*.
(TIF)

**S4 Fig. Dose-response curve for Cry5B.** Normalized animal lengths (y-axis) are plotted for each strain as a function of the dose of Cry5B in the high-throughput microscopy assay (x-axis). Strains are denoted by color. Lines extending from points represent the standard deviations from the mean responses. Statistical normalization of animal lengths is described in *Methods*.
(TIF)

**S5 Fig. Dose-response curve for derquantel.** Normalized animal lengths (y-axis) are plotted for each strain as a function of the dose of derquantel in the high-throughput microscopy assay (x-axis). Strains are denoted by color. Lines extending from points represent the standard deviations from the mean responses. Statistical normalization of animal lengths is described in *Methods*.
(TIF)

**S6 Fig. Dose-response curve for diethylcarbamazine.** Normalized animal lengths (y-axis) are plotted for each strain as a function of the dose of diethylcarbamazine in the high-throughput microscopy assay (x-axis). Strains are denoted by color. Lines extending from points represent the standard deviations from the mean responses. Statistical normalization of animal lengths is described in *Methods*.
(TIF)

**S7 Fig. Dose-response curve for emodepside.** Normalized animal lengths (y-axis) are plotted for each strain as a function of the dose of emodepside in the high-throughput microscopy assay (x-axis). Strains are denoted by color. Lines extending from points represent the standard deviations from the mean responses. Statistical normalization of animal lengths is described in *Methods*.
(TIF)

**S8 Fig. Dose-response curve for monepantel sulfide.** Normalized animal lengths (y-axis) are plotted for each strain as a function of the dose of monepantel sulfide in the high-throughput microscopy assay (x-axis). Strains are denoted by color. Lines extending from points represent the standard deviations from the mean responses. Statistical normalization of animal lengths is described in *Methods*.
(TIF)

**S9 Fig. Dose-response curve for monepantel sulfone.** Normalized animal lengths (y-axis) are plotted for each strain as a function of the dose of monepantel sulfone in the high-throughput microscopy assay (x-axis). Strains are denoted by color. Lines extending from points represent the standard deviations from the mean responses. Statistical normalization of animal lengths is described in *Methods*.
(TIF)

**S10 Fig. Dose-response curve for niridazole.** Normalized animal lengths (y-axis) are plotted for each strain as a function of the dose of niridazole in the high-throughput microscopy assay (x-axis). Strains are denoted by color. Lines extending from points represent the standard deviations from the mean responses. Statistical normalization of animal lengths is described in *Methods*.
(TIF)

**S11 Fig. Dose-response curve for oxamniquine.** Normalized animal lengths (y-axis) are plotted for each strain as a function of the dose of oxamniquine in the high-throughput microscopy assay (x-axis). Strains are denoted by color. Lines extending from points represent the standard deviations from the mean responses. Statistical normalization of animal lengths is described in *Methods*.
(TIF)

**S12 Fig. Dose-response curve for piperazine.** Normalized animal lengths (y-axis) are plotted for each strain as a function of the dose of piperazine in the high-throughput microscopy assay (x-axis). Strains are denoted by color. Lines extending from points represent the standard deviations from the mean responses. Statistical normalization of animal lengths is described in *Methods*.
(TIF)

**S13 Fig. Dose-response curve for praziquantel.** Normalized animal lengths (y-axis) are plotted for each strain as a function of the dose of praziquantel in the high-throughput microscopy assay (x-axis). Strains are denoted by color. Lines extending from points represent the standard deviations from the mean responses. Statistical normalization of animal lengths is described in *Methods*.
(TIF)

**S14 Fig. Variation in $EC_{10}$ estimates can be explained by genetic variation across strains.** For each anthelmintic, the relative potency of that anthelmintic against each strain compared to the N2 strain is shown. Solid points denote strains with significantly different relative

resistance to that anthelmintic (Student's t-test and subsequent Bonferroni correction with a $p_{adj} < 0.05$), and faded points denote strains not significantly different than the N2 strain. The broad category to which each anthelmintic belongs is denoted by the strip label for each facet. Anthelmintic drugs with undefined $EC_{10}$ estimates (estimates greater than the maximum dose to which animals were exposed) are not shown.
(TIF)

**S15 Fig. Variation in dose-response slope estimates can be explained by genetic differences among strains.** For each anthelmintic, the relative steepness of the dose-response slope inferred for that strain compared to the N2 strain is shown. Solid points denote strains with significantly different dose-response slopes (Student's t-test and subsequent Bonferroni correction with a $p_{adj} < 0.05$), and faded points denote strains without significantly different slopes than the N2 strain. The broad category to which each anthelmintic belongs is denoted by the strip label for each facet. Anthelmintic drugs with undefined slope estimates are not shown.
(TIF)

**S16 Fig. Heritability plots for anthelmintic drugs and nematicides not in the three main anthelmintic drug classes.** The broad-sense (x-axis) and narrow-sense heritability (y-axis) of normalized animal length measurements were calculated for each concentration of each nicotinic acetylcholine receptor agonist (*Methods*; *Broad-sense and narrow-sense heritability calculations*). The color of each cross corresponds to the log-transformed dose for which those calculations were performed. The horizontal line of the cross corresponds to the confidence interval of the broad-sense heritability estimate obtained by bootstrapping, and the vertical line of the cross corresponds to the standard error of the narrow-sense heritability estimate. Heritability could not be calculated for anthelmintics without an $EC_{10}$.
(TIF)

**S17 Fig. Selamectin benefits from retaining *Worm_Length* > 30 objects and the MDHD model. (A)** Distribution of animals by length (y-axis) is plotted for each strain by dose (x-axis) for selamectin in Assay C. Each data point is a single observed object. Data points are colored by worm model: L4 larval stage (purple), L2 and L3 larval stages (turquoise), L1 larval stage (green), and MDHD (pink). Average worm length by dose is denoted by the black datapoint. Selamectin high dose wells are shown for both the **(B)** *Worm_Length* > 30-pixel filter and **(C)** *Worm_Length* > 50-pixel filter. The inset shows small animals recognized by the MDHD model. Pink overlays indicate an MDHD model selected object. Animals with no overlay are filtered out during the cleaning steps. see *Methods* for details on data cleaning metrics.
(TIF)

**S18 Fig. Problematic assays and well features for downstream analysis. (A)** Distribution of animals by length (y-axis) is plotted for each strain by dose (x-axis) for niridazole in Assay H. Each data point is a single observed object. Data points are colored by worm model: L4 larval stage (purple), L2 and L3 larval stages (turquoise), L1 larval stage (green), and MDHD (pink). The black data point denotes the average worm length by dose. Well images obtained in Assay H for niridazole dose-response experiments are representative of problematic control well features caused by the presence of small animals categorized by the L1 model and L2L3 for each strain: **(B)** CB4856, **(C)** CX11314, **(D)** DL238, **(E)** JU775, **(F)** MY16, and **(G)** N2.
(TIF)

## Acknowledgments

We would like to thank members of the Andersen laboratory for their feedback and helpful comments on this manuscript. We thank the *C. elegans* Natural Diversity Resource (NSF Capacity grant 2224885) for providing us with strains for this study. We thank Joy N. Nyaanga, Timothy A. Crombie, and Samuel J. Widmayer for creating *easyXpress*, a critical software tool used in this study. We thank Samuel J. Widmayer and Timothy A. Crombie for sharing the toxicant dose-response code and workflow used in this study. We thank the following people for providing us with the following drugs: Troy Hawkins for monepantel sulfone LY33414916 and monepantel sulfide LY3348298, and Daniel Kulke for emodepside.

## Author Contributions

**Conceptualization:** Janneke Wit, Erik C. Andersen.

**Data curation:** Amanda O. Shaver.

**Formal analysis:** Amanda O. Shaver.

**Funding acquisition:** Erik C. Andersen.

**Investigation:** Janneke Wit.

**Methodology:** Janneke Wit, Clayton M. Dilks.

**Project administration:** Erik C. Andersen.

**Resources:** Hanchen Li, Raffi V. Aroian, Erik C. Andersen.

**Software:** Amanda O. Shaver, Timothy A. Crombie.

**Supervision:** Erik C. Andersen.

**Validation:** Amanda O. Shaver, Timothy A. Crombie.

**Visualization:** Amanda O. Shaver.

**Writing – original draft:** Amanda O. Shaver.

**Writing – review & editing:** Amanda O. Shaver, Janneke Wit, Clayton M. Dilks, Timothy A. Crombie, Erik C. Andersen.

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
