## [Decision Letter · Decision Letter 0]

3 Feb 2023

Dear Dr. Andersen,

Thank you very much for submitting your manuscript "Variation in anthelmintic responses are driven by genetic differences among diverse C. elegans wild strains" for consideration at PLOS Pathogens. As with all papers reviewed by the journal, your manuscript was reviewed by members of the editorial board and by several independent reviewers. The reviewers appreciated the attention to an important topic. Based on the reviews, we are likely to accept this manuscript for publication, providing that you modify the manuscript according to the review recommendations.

The reviewers all felt that the this data rich manuscript provides a valuable contribution. I also agree with their assessments. However, the reviewers identified several aspects of the manuscript where more explanation of the methods used and interpretation used is urgently needed. In addition to addressing the reviewers specific comments, I suggest paying particular attention to the following:

PloS Pathogens is not a genetics journal; a clearer explanation of the relevance of measuring heritability (H2 and h2) is needed. I suggest a clear statement of this in the introduction. In particular, the manuscript uses genomic sequence data from the C.elegans lines used to estimate SNP-based heritability. This is distinct from the more commonly understood heritability estimation methods using pedigrees or twins and also needs some explanation for non-genetics audience. I was personally surprised that accurate measures of SNP heritability can be derived from just 6 C.elegans lines - please can you provide further demonstrations that this is possible and expected from the literature?

Two reviewers felt that the 48 hr assays examining impact of drugs on worm growth may not be comparable to measures of variation in drug resistance in parasitic helminths. There was also a concern that "contractile" responses of worms to drug exposure might confound this development-based assay - please provide further justification for the approaches used.

Similarly, please further justify use of IC10, rather than the more standard IC50, or IC90. I understand that this was chosen because IC10 shows higher heritability - again, a clearer explanation of heritability is required in the introduction/results. 

Note that Rev 2 suggested replication of these results using an alternative approach to measuring resistance would improve confidence in the results. While valuable, this will not be required in the revision.

Sincerely,

Tim J.C. Anderson

Guest Editor

PLOS Pathogens

P'ng Loke

Section Editor

PLOS Pathogens

Kasturi Haldar

Editor-in-Chief

PLOS Pathogens

orcid.org/0000-0001-5065-158X

Michael Malim

Editor-in-Chief

PLOS Pathogens

orcid.org/0000-0002-7699-2064

The reviewers all felt that the this data rich manuscript provides a valuable contribution. I also agree with their assessments. However, the reviewers identified several aspects of the manuscript where more explanation of the methods used and interpretation used is urgently needed.

PloS Pathogens is not a genetics journal; a clearer explanation of the relevance of measuring heritability (H2 and h2) is needed. I suggest a clear statement of this in the introduction. In particular, the manuscript uses genomic sequence data from the C.elegans lines used to estimate SNP-based heritability. This is distinct from the more commonly understood heritability estimation methods using pedigrees or twins and also needs some explanation for non-genetics audience. I was personally surprised that accurate measures of SNP heritability can be derived from just 6 C.elegans lines - please can you provide further demonstrations that this is possible and expected from the literature?

Two reviewers felt that the 48 hr assays examining impact of drugs on growth may not be comparable to measures of variation in drug resistance in parasitic helminths. There was also a concern that "contractile" responses of worms to drug exposure might confound this development-based assay - please provide further justification for the approaches used.

Similarly, please further justify use of IC10, rather than the more standard IC50, or IC90. I understand that this was chosen because IC10 shows higher heritability - again, a clearer explanation of heritability is required in the introduction/results.

Reviewer Comments (if any, and for reference):

Reviewer's Responses to Questions

**Part I - Summary**

Reviewer #1: In this study, Dr. Shaver and colleagues authors used a panel of 6 genetically diverse C. elegans strains to perform dose-response analysis across 26 anthelmintic drugs, spanning 10 drug classes. They observed significant variation in anthelmintic responses (using microscopy-based high-throughput phenotyping assay for developmental delay in worms) of their C. elegans strains across drug classes, especially for the macrocyclic lactones and nicotinic acetylcholine receptor agonists. Finally, they quantified the heritability of the response to drugs. With this study, authors can prioritize drugs and populations of C.elegans to use for GWAS, in order to decipher the genetic basis and genes involved in drug resistance for multiple anthelmintics. Indeed, using genetically diverse populations is a key to identifying the right range of drug susceptibility and increasing the likelihood of identifying orthologous genes and mechanisms linked to drug resistance between C. elegans and parasitic nematodes.

Overall, the design of the study presented is very neat, with the appropriate controls and replications performed for each experiment, and an impressive number of individual tested. The analyses are also very robust, using the appropriate statistical tools and very well explained in the Materials & Methods section. Even if the manuscript is quite dense, it is well written and easy to understand, with great quality figures which are easily readable. One excellent point is that the authors have shared their datasets (Zenodo) and codes (Github), with is crucial in terms of rigor and reproducibility.

I have only a few minor comments and questions on the text (please see below for the details).

Reviewer #2: This paper addresses the important question of anthelmintic resistance using an elegant experimental approach that delineates the susceptibility of genetically diverse strains of C. elegans to gain insight into molecular determinants. The screen of six strains has been automated using a developmental assay, although for the sake of this relatively limited analysis it is not entirely clear what advantage that poses. This assay uses decrease in body length as a proxy for impaired development and anthelmintic effect. However, some of the compounds used will cause hypercontraction, or relaxation of the body wall muscle and this may be a confound for using body length as a metric. Furthermore, the automated approach is rather confounded by the effect of 48 hour exposure to some of the anthelmintics which may well be lethal and over this prolonged period would lead to the worms disintegrating and not amenable to a body length measurement. Indeed, this would seem to be the case and necessitated the use of filtering of the data set as indicated in the methods. The output of this study is the ability to identify drugs to prioritize for genome wide studies which ultimately will enable the identification of novel anthelmintic resistance genes. Overall the paper is well written and the work is carefully controlled.

Reviewer #3: The authors provide interesting and detailed analyses of five C. elegans natural isolates and the canonical wild type in their response to an impressive number (26) anthelmintic drugs. The study represents a lot of work and contains a lot of important information. However, I recommend that clarity be added to many of the sections to make the story easier to absorb for those who are not experts in population genetics. Detailed suggestions for improvement are listed below.

1. Abstract: It is not clear to me how the first point in the abstract (isolates respond differently to 2 of the 3 compounds tested) is different from the second point in the abstract (isolates respond differently to the compounds tested).

2. Abstract: The authors write, ‘Third, we quantified the heritability of responses to each anthelmintic and observed a significant correlation between exposure closest to the EC10 and the exposure that exhibited the most heritable responses.’. The exact meaning of ‘heritaility of responses’ is obscure to me. Is this a term common amoung population geneticists? Are the authors suggesting that the variation in the measured responses may not be heritable in the different isolates, and when tested in subsequent generations, they found that it was? Please clarify.

3. Abstract: The authors write, ‘Because genetically diverse strains displayed differential susceptibilities within and across anthelmintics, we show that C. elegans is a useful model for screening potential nematicides.’ It is not clear whether the authors mean that C. elegans is a useful pre-screening model to determine whether a candidate molecule is effective in all isolates, given their variable response, or whether they mean it is useful in some other way. Please clarify.

I hope I will understand these points once I read the body of the manuscript, but the abstract needs clarity to be understood on its own.

4. Intro: Lines 107 and 108 (and 127-128): Reads as if WGS and AR variants from CeNDR are needed to identify orthologs between elegans and parasites. Certainly, identifying orthologs (best reciprocal matches by blastp for example) can be done without CeNDR. Perhaps the authors mean something else (e.g. within a paralogous families in different species, it is often difficult to determine which single genes are the true functional ortholog of one another by sequence alone)? Please clarify.

5. Line 111. Suggestion: ‘…have repeatedly proven to translate across parasitic nematode species (refs).’

6. Line 124. Can the authors expand a little on what they mean by ‘calculate the contributions of genetics’? Naively, I would assume that statistical variation within a given strain aside, that all differences in the response of strains to a drug is due to genetics. What else is there?

7. Line 138. Explain how these 6 lines in particular were chosen.

8. Line 139. Briefly, what does ‘cleaning data’ mean?

9. Figure 1. More explanation in the legend is needed to explain step C. For example, what is OD30?

10. Line 163. I do not understand the concept of EC heritability. Please explain.

11. S1 Fig. EC10. I do not understand the math. Someone else familiar with this will have to critique.

12. Line 170-172. It would be good to have an easily interpretable figure that shows the 44 instances across 22 compounds where one isolate behaves differently than N2.

13. Line 180. Need a display item to support claims.

14. The data for the opening salvos of the paper (lines 167-191) are buried in difficult-to-absorb-quickly supplementary tables. Please have sympathy for the reader. Make it easy on us.

15. Give the reader some sense of what is meant by broad-sense and narrow-sense heritability without having to go the methods. These terms likely mean something very important to population geneticists, but mean nothing to the general reader of PLoS Pathogens. Even when I go to the methods (Lines 685-690), its all math talk without providing any intuitive meaning.

16. Lines 224-226: ‘ This heritable response indicated that genetic differences among the six strains underlie the variation in albendazole responses.’ Does one really need to do broad-sense and narrow-sense heritability calculations to come to this conclusion? The null hypothesis makes no sense- the differences in the response to the strains are not due to genetic differences’. It must be due to genetic differences, unless I am missing something.

17. Line 399. Why hypothesized?

18. Line 479. Can the authors explain how growth trends ‘can be used to assess how other nematodes might respond to the tested anthelmintic drugs’?

19. Lines 480-482. I don’t think a summary of the methodology used to be a major finding, as intended on line 477.

20. Lines 495-496. Can the authors provide proof-of-principle from their data of how using multiple isolates might help yield a candidate anthelmintic that could not be identified by using only N2?

21. After examining Figures 3a, 3b, 5a, 5b, 7a, and 7b, it is not clear that there are important/significant insights gained from the slope analyses that are not gained from EC10 analyses alone.

22. Figure 8 is a really nice summary of data presented in 3a, 3b, 5a, 5b, 7a, and 7b, along with additional data. It begs the question of whether 3a, 3b, 5a, 5b, 7a, and 7b are needed at all?

**Part II – Major Issues: Key Experiments Required for Acceptance**

Reviewer #1: None

Reviewer #2: It would lend confidence to the work if some of the key findings were reproduced using another assay of anthelmintic susceptibility. The use of body length as a read out of the impact of the compounds on development and thus a measure of sensitivity has confounds as noted above. Moreover, the analysis required what appears to be extensive filtering of the data set.

Reviewer #3: (No Response)

**Part III – Minor Issues: Editorial and Data Presentation Modifications**

Reviewer #1: ======= MAJOR COMMENTS =======

------- COMMENTS ON THE TEXT -------

* Results/Discussion:

l. 162-163 - Do the authors have a hypothesis to explain why the EC10 estimates are more heritable than the EC50 estimates?

l. 298-299: This sentence seems a bit unclear: Why do the authors not provide the h2 value for ivermectin (only the H2 value is mentioned) and the H2 value for doramectin (only the h2 value is mentioned)?

* Figures:

- Dose-Response curves Figures - legend: For each strain and each drug dose, how many worms have been treated? Adding N values to the legends of the dose-response curves would be useful.

* Tables:

- Table 1: Perhaps worth adding a column to the Table 1 mentioning the range of doses used for each drugs in this study.

* Materials & Methods:

§ C. elegans strain selection and maintenance: What is the percentage of genetic divergence between the different stains chosen? Perhaps good to mention in this paragraph. This will clarify, for the reader, why these specific 6 C. elegans strains have been chosen, especially if they are very divergent genetically between each other.

§ High-throughput anthelmintic dose-response assay: In this experimental design, the worms were exposed to various doses of drugs during 48h. Why has this specific treatment/incubation time have been chosen? Have the authors observed similar results with a shorter treatment time (24h)? Would it be possible to treat worms for 24h and then culture them for 24h without drug and assess their phenotypic response? I suspect it would be difficult, as these worms are tiny, but it would better mimic what could happen in-vivo with parasitic nematodes, when hosts are treated with anthelmintics. Perhaps good here to justify briefly why this specific exposure time was chosen.

----- COMMENTS ON THE REFERENCES -----

None

Reviewer #2: Why would EC10 estimates be more heritable than EC50 estimates? A comment of this would be helpful.

Its not helpful having so much of the data as supplementary information.

Fig 8. How was relative potency calculated?

line 458 – only one phenotype was measured so this is a bit of an overstatement

line 476- analysis did not measure growth rates- it only measured an end point

Add information on the mode of action of the different compounds e.g., add to table 1

line 616- what is this 'debris'? dead worms? how can this be excluded if this is the case?

line 624- 30 worms were the starting point but less than 3 in a well excluded? this suggests very large variation in the data

Reviewer #3: (No Response)

PLOS authors have the option to publish the peer review history of their article (what does this mean?). If published, this will include your full peer review and any attached files.

Reviewer #1: No

Reviewer #2: No

Reviewer #3: No

Figure Files:

Data Requirements:

Reproducibility:

References:

---

## [Editor Report · Decision Letter 1]

8 Mar 2023

Dear Dr. Andersen,

We are pleased to inform you that your manuscript 'Variation in anthelmintic responses are driven by genetic differences among diverse C. elegans wild strains' has been provisionally accepted for publication in PLOS Pathogens. Thank you for your thorough response to the reviewers critiques, and for making this paper more accessible to a non-genetics audience.

Best regards,

Tim J.C. Anderson

Guest Editor

PLOS Pathogens

P'ng Loke

Section Editor

PLOS Pathogens

Kasturi Haldar

Editor-in-Chief

PLOS Pathogens

orcid.org/0000-0001-5065-158X

Michael Malim

Editor-in-Chief

PLOS Pathogens

orcid.org/0000-0002-7699-2064
---

## [Editor Report · Acceptance letter]

29 Mar 2023

Dear Dr. Andersen,

We are delighted to inform you that your manuscript, "Variation in anthelmintic responses are driven by genetic differences among diverse *C. elegans* wild strains," has been formally accepted for publication in PLOS Pathogens.

Best regards,

Kasturi Haldar

Editor-in-Chief

PLOS Pathogens

orcid.org/0000-0001-5065-158X

Michael Malim

Editor-in-Chief

PLOS Pathogens

orcid.org/0000-0002-7699-2064